# Cangrelor Use in Routine Practice: A Two-Center Experience

**DOI:** 10.3390/jcm10132829

**Published:** 2021-06-26

**Authors:** Niels M. R. van der Sangen, Ho Yee Cheung, Niels J. W. Verouden, Yolande Appelman, Marcel A. M. Beijk, Bimmer E. P. M. Claessen, Ronak Delewi, Paul Knaapen, Jorrit S. Lemkes, Alexander Nap, M. Marije Vis, Wouter J. Kikkert, José P. S. Henriques

**Affiliations:** 1Department of Cardiology, Amsterdam UMC, University of Amsterdam, Amsterdam Cardiovascular Sciences, 1105 AZ Amsterdam, The Netherlands; n.m.r.vandersangen@amsterdamumc.nl (N.M.R.v.d.S.); h.y.cheung@amsterdamumc.nl (H.Y.C.); m.a.beijk@amsterdamumc.nl (M.A.M.B.); b.e.claessen@amsterdamumc.nl (B.E.P.M.C.); r.delewi@amsterdamumc.nl (R.D.); m.m.vis@amsterdamumc.nl (M.M.V.); w.j.kikkert@olvg.nl (W.J.K.); 2Department of Cardiology, Amsterdam UMC, VU University, Amsterdam Cardiovascular Sciences, 1081 HV Amsterdam, The Netherlands; c.verouden@amsterdamumc.nl (N.J.W.V.); y.appelman@amsterdamumc.nl (Y.A.); p.knaapen@amsterdamumc.nl (P.K.); j.lemkes@amsterdamumc.nl (J.S.L.); a.nap@amsterdamumc.nl (A.N.); 3Department of Cardiology, Onze Lieve Vrouwe Gasthuis, 1091 AC Amsterdam, The Netherlands

**Keywords:** cangrelor, antiplatelet therapy, percutaneous coronary intervention

## Abstract

Cangrelor is the first and only intravenous P2Y_12_-inhibitor and is indicated when (timely) administration of an oral P2Y_12_ inhibitor is not feasible in patients undergoing percutaneous coronary intervention (PCI). Our study evaluated the first years of cangrelor use in two Dutch tertiary care centers. Cangrelor-treated patients were identified using a data-mining algorithm. The cumulative incidences of all-cause death, myocardial infarction, definite stent thrombosis and major bleeding at 48 h and 30 days were assessed using Kaplan–Meier estimates. Predictors of 30-day mortality were identified using uni- and multivariable Cox regression models. Between March 2015 and April 2021, 146 patients (median age 63.7 years, 75.3% men) were treated with cangrelor. Cangrelor was primarily used in ST-segment elevation myocardial infarction (STEMI) patients (84.2%). Approximately half required cardiopulmonary resuscitation (54.8%) or mechanical ventilation (48.6%). The cumulative incidence of all-cause death was 11.0% and 25.3% at 48 h and 30 days, respectively. Two cases (1.7%) of definite stent thrombosis, both resulting in myocardial infarction, occurred within 30 days, but after 48 h. No other cases of recurrent myocardial infarction transpired within 30 days. Major bleeding occurred in 5.6% and 12.5% of patients within 48 h and 30 days, respectively. Cardiac arrest at presentation was an independent predictor of 30-day mortality (adjusted hazard ratio 5.20, 95%-CI: 2.10–12.9, *p* < 0.01). Conclusively, cangrelor was used almost exclusively in STEMI patients undergoing PCI. Even though cangrelor was used in high-risk patients, its use was associated with a low rate of stent thrombosis.

## 1. Introduction

In 2015 cangrelor became the first intravenous P2Y_12_-inhibitor approved for clinical use by the European Medicines Agency (EMA). Cangrelor can be used during percutaneous coronary intervention (PCI), alongside aspirin, in patients who have not received an oral P2Y_12_-inhibitor prior to PCI and when oral P2Y_12_-inhibition is not feasible [1,2]. As opposed to oral P2Y_12_-inhibtors, cangrelor has the pharmacokinetic advantage of reducing platelet reactivity within minutes of administration and returning normal platelet function within 30–60 min after treatment discontinuation due to a relatively short half-life [3]. 

The efficacy and safety of cangrelor was first evaluated in the three Cangrelor versus Standard Therapy to Achieve Optimal Management of Platelet Inhibition (CHAMPION) randomized clinical trials [4,5,6]. Pooled data of these trials (n = 24,910) showed that the use of cangrelor was associated with a reduction in peri-procedural ischemic complications (i.e., all-cause death, myocardial infarction, ischemia-driven revascularization or stent thrombosis within 48 h) and an increase in minor (but not major) bleeding events as compared to clopidogrel [7]. Ever since cangrelor gained market authorization, there has been limited data on its use and outcomes in cangrelor-treated patients in routine practice. Therefore, the objective of this study was to describe (i) patient and procedural characteristics, (ii) clinical outcomes and (iii) the transition to oral P2Y_12_-inhibitors of cangrelor-treated patients undergoing coronary angiography or PCI in two Dutch tertiary PCI centers during the first years of routine cangrelor use. 

## 2. Materials and Methods

### 2.1. Source Population and Standard Procedures 

The data analyzed in this manuscript were obtained from cangrelor-treated patients undergoing coronary angiography with intent of PCI in the two affiliated hospitals of the Amsterdam UMC, University Medical Centers. Both hospitals are high-volume, tertiary PCI centers situated in Amsterdam, the Netherlands. Patients with ST-segment elevation myocardial infarction (STEMI) were routinely pretreated with 160 mg aspirin (or 500 mg aspirin i.v. in unconscious or noxious patients), 180 mg ticagrelor or 60 mg prasugrel and 5000 IU of unfractionated heparin as soon as the diagnosis was established. In patients with non-ST-segment elevation myocardial infarction (NSTEMI) and unstable angina a similar loading dose of aspirin and ticagrelor or prasugrel was combined with 2.5 mg fondaparinux s.c. once daily until PCI. In patients using oral anticoagulants ticagrelor or prasugrel was substituted for clopidogrel. Patients undergoing PCI for chronic coronary syndrome were started on 75 mg clopidogrel and 100 mg aspirin at least 3 days prior to the procedure. All patients received a weight-adjusted dose of unfractionated heparin at the catheterization laboratory and an additional dose after 60 min if the activated clotting time fell below 250 s. In both centers, cangrelor was indicated if pretreatment with an oral P2Y_12_-inhbitor was not feasible or desirable and PCI could not be postponed. Cangrelor was administered as a bolus of 30 µg/kg followed by an infusion of 4 µg/kg/min for at least 2 h or the duration of the procedure, whichever lasted longer. Patients received a loading dose of an oral P2Y_12_-inhibitor directly after completion of cangrelor infusion.

### 2.2. Patient Identification Method and Data Collection

A Boolean retrieval query was developed in order to identify cangrelor-treated patients based on data from their electronic health record. This query was developed using a graphic user interface data-mining algorithm with text-mining features (CTcue, version 2.1.13, Amsterdam, The Netherlands). Both structured and unstructured data from electronic health records, such as drug prescriptions and procedural reports, were integrally searched using the data-mining algorithm with the following search terms: ‘Cangrelor’ and ‘Kengrexal’. Additional filters were applied to exclude patients under the age of 18 and patients treated with cangrelor before 23 March 2015, the date of the EMA marketing authorization. The results from the automated search were manually verified. Patients were given a four-week period to object to the (re)use of their clinical care data. Demographic, clinical and procedural characteristics were collected through chart review.

### 2.3. Clinical Outcomes

The primary endpoints were all-cause death, myocardial infraction, definite stent thrombosis and major bleeding at 48 h and 30 days. Myocardial infarction was defined in accordance with the fourth universal definition of myocardial infarction [8]. Definite stent thrombosis was defined according to the Academic Research Consortium (ARC) definition [9]. Bleeding events were classified according to the Bleeding Academic Research Consortium (BARC) bleeding definition [10]. BARC type 3 to 5 bleeding was considered major, while BARC type 2 bleeding was considered minor. All primary endpoints were adjudicated by two authors (N.M.R.v.d.S. and H.Y.C.). Timing of oral loading dose was subdivided in pre-, during or post-PCI. Furthermore, the choice for a specific P2Y_12_-inhibitor was recorded (clopidogrel, ticagrelor or prasugrel). In case of an oral P2Y_12_-inhibitor switch, only the first choice was reported. The possible administration of an additional, second loading dose (‘reloading’) was also noted.

### 2.4. Statistical Methods

The primary objective of our study was descriptive in nature; therefore, no formal sample size calculation was performed. Continuous variables were reported as mean (standard deviation) or median (interquartile range) and categorical variables were reported as number of patients (percentage). Cumulative incidences of the primary endpoints were assessed using Kaplan–Meier estimates at 48 h and 30 days. Data from patients who did not have a primary endpoint event between hospital admission and 30 days were censored at the time of death, the time of last follow-up or 30 days, whichever came first. Cox proportional hazard models were used to identify (potential) predictors of 30-day mortality. Uni- and multivariable models were used. In the latter, age, sex and all statistically significant predictors from the univariate analyses were included. A *p*-value of <0.05 was considered statistically significant. Statistical analyses were performed using SPSS version 26 (SPSS Inc., Chicago, IL, USA) and R studio version 3.6.1 (Vienna, Austria).

## 3. Results

### 3.1. Patient and Procedural Characteristics

From March 2015 through April 2021, 166 patients were identified by our data-mining algorithm. A total of 150 patients were considered to be cangrelor-treated after manual validation of the search results. In 16 patients, cangrelor treatment was considered but ultimately not administered. Four patients objected to the (re)use of their clinical care data. Hence, 146 patients were included in the data analysis (Figure 1).

Baseline and procedural characteristics of cangrelor-treated patients are presented in Table 1 and Table 2. Cangrelor-treated patients had a mean age of 63.7 years and were most often male (75.3%). Cangrelor was primarily used in STEMI patients (84.2%) with high-risk features, such as cardiac arrest (54.8%), mechanical ventilation (48.6%) and/or cardiogenic shock (45.2%). Eighteen patients (12.3%) received mechanical circulatory support during admission. Radial access was used slightly more than femoral access (54.8% vs. 45.2%). PCI was performed in 140 patients (94.3%) using drug-eluting stents in all but eight cases. Twenty-six patients underwent multi-vessel PCI (18.6%) and thrombectomy was performed in 28 patients (20.0%). All patients started cangrelor right before or during coronary angiography. The vast majority of patients underwent coronary angiography within the first 24 h of hospital admission (93.8%).

Almost all patients received unfractionated heparin during coronary angiography (97.9%). One patient (0.7%) received glycoprotein IIb/IIIa bailout therapy using abciximab alongside cangrelor. Pretreatment with an oral P2Y_12_-inhibitor was initiated before PCI in approximately one quarter of cases (26.6%), while 2.9% and 70.5% of cases received their initial loading dose during or after PCI. Five patients (3.6%) received an additional P2Y_12_-inhibitor loading dose (‘reloading’) due to vomiting or nausea after being pretreated before the procedure. Ticagrelor was the most commonly used P2Y_12_-inhibitor (89.2%), although some patients were treated with clopidogrel (10.8%). 

### 3.2. Clinical Outcomes

Clinical outcomes at 48 h and 30 days follow-up are presented in Table 3 and Figure 2. The incidence of all-cause death was 11.0% and 25.3% at 48 h and 30 days, respectively. All deaths within 48 h had a cardiovascular cause. One patient died due to intracranial bleeding 19 days after cangrelor treatment. The median time until death was 3 days (IQR 1-9). Definite stent thrombosis, resulting in myocardial infarction (type 4b), occurred in two separate cases (1.7%). The first case of stent thrombosis (complete stent occlusion) occurred in a 51-year-old male 10 days after presentation with cardiac arrest and anterior STEMI. The patient was initially treated with a drug-eluting stent in the proximal left anterior descending (LAD) coronary artery and was transitioned to ticagrelor in conjunction with aspirin. Upon re-angiography at day 10 indicated by refractory angina and ischemic electrocardiogram changes, optical coherence tomography (OCT) revealed stent malapposition and a non-obstructing edge dissection proximal of the stent. Balloon dilatation and thrombosuction were ultimately successfully. The second case of stent thrombosis occurred at day 21 of admission (and 8 days after cangrelor therapy) in a 71-year-old male admitted for ischemic heart failure and (in-hospital) cardiac arrest. His medical history included active colorectal carcinoma without signs of metastases for which a hemicolectomy was planned. The patient was initially on clopidogrel and aspirin, but was switched to ticagrelor and aspirin. Re-angiography indicated by ventricular fibrillation showed stent thrombosis (complete stent occlusion) of the stent implanted days before in the mid LAD. Multiple attempts of wire passage were unsuccessful and the patient expired four days later. No other cases of myocardial infarction transpired within 30 days. Major or minor bleeding occurred in 13.8% and 22.0% of patients within 48 h and 30 days, respectively. The estimated cumulative incidence of major bleeding was 5.6% and 12.5% at 48 h and 30 days. Out of the eight major bleeding events within 48 h, two were access site-related, two were device-related, one was gastrointestinal, one was intracranial, one was intrathoracic and one was presumed to be based on a significant drop in hemoglobin, but with an unidentified source. In total, four gastrointestinal, four device-related, three access site-related, two intracranial, one intrathoracic and three presumed major bleeding events occurred within 30 days of hospital admission. 

Results of the Cox proportional hazard analyses of predictors for 30-day mortality are shown in Table 4. Cardiac arrest at presentation was a statistically significant predictor for 30-day mortality (unadjusted hazard ratio 5.00, 95%-CI: 2.08–12.0, *p* < 0.01). After adjustment for age and sex in the multivariable model, cardiac arrest remained an independent predictor of 30-day mortality (adjusted hazard ratio 5.20, 95%-CI: 2.10–12.9, *p* < 0.01).

## 4. Discussion

The most important findings of the current study are as follows: (i) intravenous P2Y_12_ inhibition with cangrelor is used in a small proportion of patients undergoing invasive cardiac procedures, (ii) patients receiving cangrelor are generally high-risk patients (including patients after cardiac arrest, on mechanical ventilation and/or in cardiogenic shock), and (iii) patients receiving cangrelor had low rates of stent thrombosis and acceptable bleeding rates, given their high-risk profile.

In our study cangrelor was primarily used in high-risk STEMI patients. These findings are consistent with previous work in other registries [11,12]. Grimfjärd et al. already showed that cangrelor was used almost exclusively in STEMI patients (98.2% of all cases), often presenting with cardiac arrest (17.8%), in the Swedish Coronary Angiography and Angioplasty Registry (SCAAR) between 2016 and 2018 [11]. Conversely, in the original CHAMPION trials, only 11.6% of the pooled study population consisted of STEMI patients [7]. Notably, patients with cardiogenic shock or cardiac arrest were not included in the CHAMPION trials, so the efficacy and safety of cangrelor in these patients is relatively unknown.

### 4.1. Ischemic and Bleeding Complications

It has been well established that high-risk patients, mainly cardiac arrest or cardiogenic shock patients, have an excess risk of early ischemic complications, such as stent thrombosis [13]. High on-treatment platelet reactivity has been reported up to several hours after an oral P2Y_12_-inhibitor loading dose (even with ticagrelor or prasugrel) and there is a high degree of inter-individual variability [14,15]. The delayed onset of action has been attributed to factors such as reduced drug bioavailability due to impaired drug absorption (e.g., in cardiogenic shock patients) [16,17]. Organ hypoperfusion also effects the hepatic cytochrome P450 enzyme system, which is responsible for the biotransformation of clopidogrel and prasugrel (both prodrugs) to their active metabolites [18]. Likewise, iatrogenic factors such as medically induced hypothermia (after cardiac arrest) and morphine use can result in an attenuated antiplatelet effect [19,20,21,22]. Cangrelor circumvents some of the contribution factors to the excess ischemic risk. In the present study, no cases of stent thrombosis occurred in the first 48 h after PCI, suggesting that platelet inhibition during this period was adequate. Similarly, in a retrospective evaluation of cangrelor use and peri-procedural outcomes in patients in clinical shock, Vaduganathan et al. reported that only one out of 147 (0.7%) cangrelor-treated patients had a stent thrombosis within 48 h [12]. Additionally, in the SCAAR registry only 0.7% of all cangrelor-treated STEMI patients had a stent thrombosis within 30 days [11]. Although the reported rate of stent thrombosis in these patients has been reassuring, the balance between ischemic protection and additional bleeding risk remains precarious. Critically ill patients are at increased risk of (major) bleeding, partially due to frequent use of femoral access, mechanical circulatory support devices and gastro-intestinal hypoperfusion resulting in stress ulcers. Illustratively, Vaduganathan et al. reported that Global Use of Strategies to Open Occluded Coronary Arteries (GUSTO) moderate or mild bleeding occurred in 29% of cangrelor-treated patients in clinical shock within 48 h [12]. The bleeding rate in clinical shock patients was approximately twice as high as in hemodynamically stable patients [12]. Strategies to minimizing bleeding events in this vulnerable population might be increasing use of radial access and avoiding concurrent administration of glycoprotein IIb/IIIb inhibitors [12]. In line with other observational studies, cangrelor was only rarely combined with glycoprotein IIb/IIIa inhibitors in our cohort [11,12]. Post hoc analyses in CHAMPION already showed that cangrelor’s efficacy in patients undergoing PCI was maintained irrespective of glycoprotein IIb/IIIa inhibitor use and that cangrelor alone was associated with a similar ischemic risk and lower risk-adjusted bleeding risk compared with clopidogrel combined with glycoprotein IIb/IIIa inhibitors [23,24].

### 4.2. Timing and Choice of Oral P2Y_12_-Inhibititors

The efficacy and safety of cangrelor was originally compared with clopidogrel and not with the more potent P2Y_12_-inhibitors ticagrelor or prasugrel [7], which are the standard-of-care according to current guidelines [1,25]. In two out of the three CHAMPION trials, pretreatment (before PCI) with clopidogrel was not required, although this is common practice in the Netherlands [5,6]. In the current study, one quarter of patients received a P2Y_12_-inhibitor pre-PCI, although cangrelor is officially only indicated in patients who have not received an oral P2Y_12_-inhibitor prior to PCI. Importantly, the Platelet Inhibition With Cangrelor and Crushed Ticagrelor in STEMI Patients Undergoing Primary Percutaneous Coronary Intervention (CANTIC) study showed that (crushed) ticagrelor could be concomitantly administered with cangrelor without any apparent drug–drug interaction [26]. Other studies have shown that platelet inhibition during the transition from cangrelor to ticagrelor is maintained [27,28]. Conversely, the transition from cangrelor to clopidogrel and prasugrel is hampered by the competitive pharmacodynamics effects of these agents [29,30,31]. Therefore, oral loading with clopidogrel or prasugrel should take place only after the end of cangrelor infusion, possibly leaving a small, but critical, time window for inappropriate platelet aggregation [28].

### 4.3. Future Directions

Balancing ischemic and bleeding risk in high-risk patients, such as patients in cardiogenic shock or after cardiac arrest, remains a major challenge in clinical practice. At present, these patients are most often treated with cangrelor followed by ticagrelor (administered orally or through a nasogastric tube). Future studies should further investigate the added value of cangrelor in these high-risk patients, preferably in randomized controlled trials. Currently, the Dual Antiplatelet Therapy For Shock Patients With Acute Myocardial Infarction (DAPT-SHOCK-AMI) trial (ClinicalTrials.gov Identifier: NCT03551964) is comparing cangrelor and ticagrelor (administered orally or through a nasogastric tube) in 304 patients with acute myocardial infarction complicated by cardiogenic shock and treated with primary PCI. Hopefully, this trial will provide more insight into the net clinical effect (the combination of major adverse cardiac events and bleeding events) of cangrelor in high-risk patients. Similarly, the Cangrelor in Comatose Survivors of OHCA Undergoing Primary PCI (Cangrelor OHCA) trail (ClinicalTrials.gov Identifier: NCT04005729) is investigating if a 4 h continuous infusion of cangrelor at the start of primary PCI immediately and effectively suppresses platelet activity in comatose survivors of cardiac arrest. Results of these and other trials are eagerly awaited.

### 4.4. Limitatioins

Our study has several limitations. Our study is descriptive in nature and there is no formal control group for the cangrelor-treated patients. Data were collected retrospectively, leaving open the possibility of information bias, especially in patients who died soon after being admitted. Furthermore, we only studied the rate of definite stent thrombosis, which requires angiographic or pathological confirmation and might therefore underestimate the true stent thrombosis rate. Moreover, the sample size was limited, therefore the study was not powered to investigate predictors of 30-day mortality and possible subgroup differences (e.g., sex differences). Finally, the study was conducted in tertiary PCI centers and results may therefore not be generalizable to all PCI centers.

## 5. Conclusions

In routine practice, cangrelor was used almost exclusively in STEMI patients undergoing PCI. Even though cangrelor was used in high-risk patients, often requiring cardiopulmonary resuscitation or mechanical ventilation, its use was associated with a low rate of stent thrombosis.

## Figures and Tables

**Figure 1 jcm-10-02829-f001:**
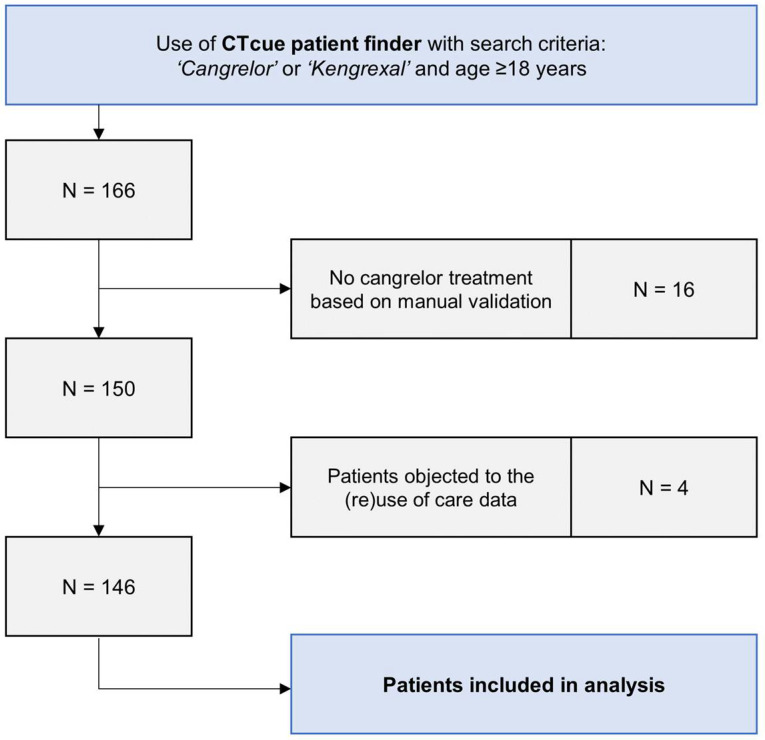
Flowchart of cangrelor-treated patients in both participating centers.

**Figure 2 jcm-10-02829-f002:**
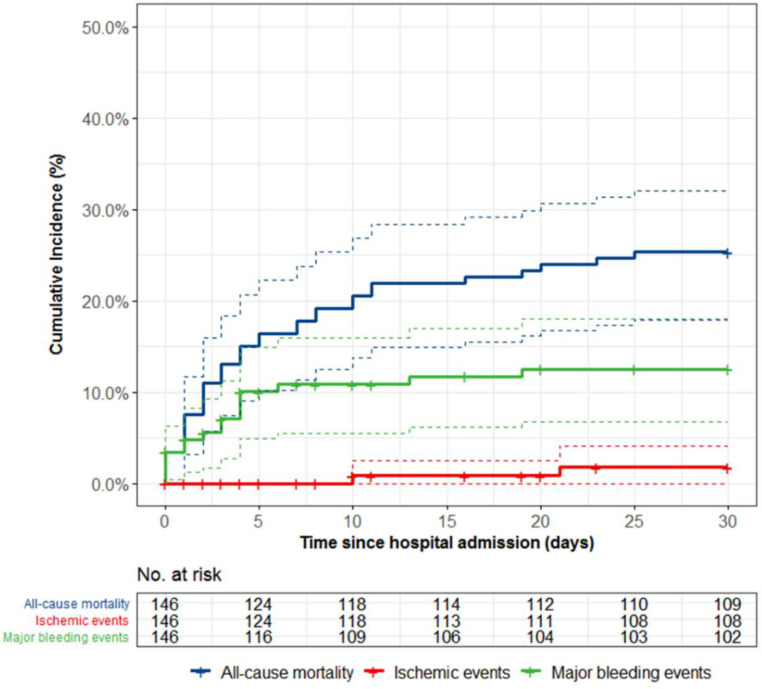
Kaplan–Meier curve including 95% confidence interval (dashed lines) for mortality, ischemic and bleeding events in cangrelor-treated patients. Ischemic events were defined as stent thrombosis or (recurrent) myocardial infarction. Major bleeding events were defined as BARC type 3 to 5 bleeding.

**Table 1 jcm-10-02829-t001:** Baseline characteristics and clinical presentation of cangrelor-treated patients.

	Cangrelor-Treated Patients (*n* = 146)
**Demographic Characteristics**	
Age (years)	63.7 ± 11.7
Male, no./total no. (%)	110/146 (75.3%)
Body mass index (kg/m^2^) *	26.8 ± 3.7
Hypertension, no./total no. (%)	50/140 (35.7%)
Dyslipidemia, no./total no. (%)	35/104 (33.7%)
Current smokers, no./total no. (%)	46/118 (39.0%)
Diabetes mellitus, no./total no. (%) Insulin-dependent diabetes, no./total no. (%)	23/141 (16.3%) 7/141 (5.0%)
Positive family history, no./total no. (%) †	35/87 (40.2%)
Chronic kidney disease, no/total no. (%) ‡	6/143 (4.2%)
Prior myocardial infarction, no./total no. (%)	18/143 (12.6%)
Prior PCI, no./total no. (%)	19/143 (13.3%)
Prior CABG, no./total no. (%)	6/143 (4.2%)
Prior stroke or TIA, no./total no. (%)	9/143 (6.3%)
**Clinical presentation**	
Diagnosis at presentation, no./total no. (%) STEMI NSTEMI Unstable angina Chronic coronary syndrome Other diagnosis	123/146 (84.2%) 10/146 (6.8%) 2/146 (1.4%) 4/146 (2.7%) 7/146 (4.8%)
Killip classification, no./total no. (%) Class I Class II Class III Class IV	62/146 (42.5%) 14/146 (9.6%) 4/146 (2.7%) 66/146 (45.2%)
Cardiac arrest at presentation, no./total no. (%)	80/146 (54.8%)
Mechanical ventilation, no./total no. (%)	71/146 (48.6%)
Mechanical circulatory support, no./total no. (%) §	18/146 (12.3%)

Plus–minus values are mean ± standard deviation or number of patients (percentage). CABG denotes coronary artery bypass grafting, NSTEMI non–ST-segment elevation myocardial infarction, PCI percutaneous coronary intervention, STEMI ST-segment elevation myocardial infarction, TIA transient ischemic attack. * Body mass index was calculated by dividing weight in kilograms by the square of the height in meters and was available in 116/146 patients. † Positive family history was defined as a (fatal) cardiovascular event and/or an established diagnosis of cardiovascular disease in a first-degree relatives before 65 years of age. ‡ Chronic kidney disease was defined as an estimated glomerular filtration rate < 60 mL/min/1.73 m^2^. § Mechanical circulatory support was defined as the use of extracorporeal membrane oxygenation (ECMO), Impella or an intra-aortic balloon pump (IABP).

**Table 2 jcm-10-02829-t002:** Procedural characteristics of cangrelor-treated patients.

	Cangrelor-Treated Patients (*n* = 146)
**Procedural Characteristics**	
Procedural length (min)	44 (34–62)
Access route, no./total no. (%) Radial access Femoral access	80/146 (54.8%) 66/146 (45.2%)
PCI, no./total no. (%) Drug-eluting stent Balloon angioplasty	140/146 (95.9%) 132/140 (94.3%) 8/140 (5.7%)
Target vessel, no./total no. (%) * Left main Left anterior descending Left circumflex artery Ramus intermedius Right coronary artery Saphenous vein graft	18/140 (12.9%) 76/140 (54.3%) 25/140 (17.9%) 10/140 (7.1%) 45/140 (32.1%) 1/140 (0.7%)
Multi-vessel PCI, no./total no. (%)	26/140 (18.6%)
Aspiration thrombectomy, no./total no. (%)	28/140 (20.0%)
Unsuccessful or no PCI, no./total no. (%)	6/146 (4.1%)
**Procedural medication**	
Aspirin loading dose, no./total no. (%)	138/146 (94.5%)
P2Y_12_-inhibitor loading dose, no./total no. (%)	139/146 (95.2%)
Type of oral P2Y_12_-inhibitor, no./total no. (%) Clopidogrel Prasugrel Ticagrelor	15/139 (10.8%) 0/139 (0.0%) 124/139 (89.2%)
Timing of P2Y_12_-inhibitor loading dose, no./total no. (%) Pre-PCI During PCI Post-PCI	37/139 (26.6%) 4/139 (2.9%) 98/139 (70.5%)
P2Y_12_-inhibitor reloading, no./total no. (%) †	5/139 (3.6%)
Oral anticoagulants use, no./total no. (%) Vitamin K antagonist Direct oral anticoagulant	5/146 (3.4%) 6/146 (4.1%)
Peri-procedural anticoagulants, no./total no. (%) Unfractionated heparin Glycoprotein IIb/IIIa inhibitors	143/146 (97.9%) 1/146 (0.7%)

Plus–minus values are median (25th–75th interquartile range) or number of patients (percentage). PCI denotes percutaneous coronary intervention. * Right coronary artery includes right posterolateral branch, left anterior descending includes diagonal branches, left circumflex artery includes marginal branches. † P2Y_12_-inhibitor reloading was defined as receiving an additional (i.e., second) loading dose after the initial loading dose during the same hospital admission.

**Table 3 jcm-10-02829-t003:** Cumulative incidence of ischemic and bleeding events at 48 h and 30 days.

	Cangrelor-Treated Patients (*n* = 146)	
	48 h	30 Days
All-cause mortality, no. of patients (%)	16 (11.0%)	37 (25.3%)
Myocardial infarction, no. of patients (%)	0 (0.0%)	2 (1.7%)
Definite stent thrombosis, no. of patients (%)	0 (0.0%)	2 (1.7%)
Major bleeding, no. of patients (%) Access site bleeding	8 (5.6%) 2 (25.0%) *	17 (12.5%) 3 (17.6%) *
Major or minor bleeding, no. of patients (%) Access site bleeding	20 (13.8%) 4 (20.0%) *	31 (22.0%) 5 (16.1%) *

Percentages are Kaplan–Meier estimates of the cumulative incidence of the clinical endpoints. * Percentage of bleeding events defined as access site bleeding relative to all bleeding events.

**Table 4 jcm-10-02829-t004:** Predictors of 30-day mortality in cangrelor-treated patients.

	Univariate Analysis		Multivariable Analysis	
	HR (95%-CI)	*p*-Value	HR (95%-CI)	*p*-Value
Age per year *	1.00 (0.97–1.02)	0.74	1.01 (0.98–1.04)	0.71
Male sex	1.41 (0.62–3.21)	0.41	0.92 (0.40–2.14)	0.85
Hypertension	1.56 (0.80–3.04)	0.19		
Diabetes mellitus	0.80 (0.31–2.05)	0.64		
Current smoker	0.82 (0.37–1.84)	0.63		
Pervious MI	0.61 (0.19–2.00)	0.61		
Previous revascularization †	0.93 (0.39–2.24)	0.87		
STEMI at presentation	1.63 (0.58–4.59)	0.36		
Cardiac arrest at presentation	5.00 (2.08–12.0)	<0.01	5.20 (2.10–12.9)	<0.01

CI denotes confidence interval, HR hazard ratio, MI myocardial infarction, PCI percutaneous coronary intervention, STEMI ST-segment elevation myocardial infarction. * Note that the HR corresponds to a unit increase in the explanatory variable. † Revascularization was defined as previous PCI and/or CABG.

## Data Availability

Data are available upon reasonable request.

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
