# Peer review of "Cangrelor Use in Routine Practice: A Two-Center Experience"

_jcm, 2021, doi:10.3390/jcm10132829_

Round 1

Reviewer 1 Report

Interesting description of the use of Cangrelor. The obvious limitations (small number of patients, no control etc are well noted by the reviewers) 

just 2 minor issues:

L 39 should be "is the only....."

L61 not clear what the authors mean with patients undergoing CAG or PCI. Probably they mean PCI only.

I find the bleeding rate rather high and I am worried about this 1 patient with ICH. Please address this in the discussion and report rates (especially ICH) with the use of cangrelor

Reviewer 2 Report

This is a retrospective, registry based, non-comparative study, conducted in two centers in the Netherlands, aimed at describing the management indications and problems of a relatively new anti-platelet agent belonging to the P2Y12 inhibitors class. The agent is available only in the intravenous form and is  characterized by a very fast onset of its anti-aggregation activity and a fast disappearance rate due to short half-life. Cangrelor was used by Authors only in STEMI patients, especially if oral agents could not be used owing to severe conditions, and was associated to other antithrombotic medications as low dose Aspirin and variable use of low molecular heparin or fandaparinux. Authors report these protocols in the text. Authors recognize the weakpoints of their study: the relatively low number of patients (compensated by their severe conditions), and especially the lack of a control group that prevents an accurate  comparison, especially because patients received other antithrombotic agents.

However, Authors believe that early Cangrelor treatment may have reduced the incidence especially of early coronary re-occlusion, an effect that is credible given the characteristics of the agent.
